# A Training-Free Foreground–Background Separation-Based Wire Extraction Method for Large-Format Transmission Line Images

**DOI:** 10.3390/s25216636

**Published:** 2025-10-29

**Authors:** Ning Liu, Yuncan Bai, Jingru Liu, Xuan Ma, Yueming Huang, Yurong Guo, Zehua Ren

**Affiliations:** 1State Grid Electric Power Space Technology Company Limited, Beijing 102209, China; liuning1396652025@163.com (N.L.); liujingru_186@163.com (J.L.); marine_mxx@163.com (X.M.); huang_yueming0327@163.com (Y.H.); 2United Laboratory of Digital Grid & Power Space Technology, Beijing 102209, China; 3Yanzhao Electric Power Laboratory, North China Electric Power University, Baoding 071003, China; 4Department of Electronic and Communication Engineering, North China Electric Power University, Baoding 071003, China; renzehua@ncepu.edu.cn

**Keywords:** deep power vision technology, large-format images, transmission line inspection, wire extraction

## Abstract

With the rapid development of smart grids, deep power vision technologies are playing a vital role in monitoring the condition of transmission lines. In particular, for high-resolution and large-format transmission line images acquired during routine inspections, accurate extraction of transmission wires is crucial for efficient and accurate subsequent defect detection. In this paper, we propose a training-free (i.e., requiring no task-specific training or annotated datasets for wire extraction) wire extraction method specifically designed for large-scale transmission line images with complex backgrounds. The core idea is to leverage depth estimation maps to enhance the separation between foreground wires and complex backgrounds. This improved separability enables robust identification of slender wire structures in visually cluttered scenes. Building on this, a line segment structure-based method is developed, which identifies wire regions by detecting horizontally oriented linear features while effectively suppressing background interference. Unlike deep learning-based methods, the proposed method is training-free and dataset-independent. Experimental results show that our method effectively addresses background complexity and computational overhead in large-scale transmission line image processing.

## 1. Introduction

With the continued advancement of smart grid development, the power system is facing increasingly stringent demands for intelligent operation monitoring, fault prediction, and maintenance response. As critical infrastructure for electricity transmission, transmission lines serve as essential links between the power supply and load sides of the grid. Their operational security and stability are directly tied to the overall reliability of the power system and the continuity of the electricity supply. In this context, the integration of deep power vision technologies provides new opportunities for enhancing the intelligence level of transmission line monitoring, enabling more accurate perception and analysis of their operating states. Over prolonged periods of service, transmission lines are prone to various structural defects—such as corrosion and aging of metallic components, broken or loosened strands, wire slack, and foreign object attachments—which can lead to abnormal power transmission and, in severe cases, result in widespread outages or equipment damage. Early detection and timely mitigation of such hidden hazards are vital for enhancing the resilience of power system operations and improving accident prevention capabilities. Therefore, achieving efficient, accurate, and real-time monitoring of transmission line conditions has become a fundamental requirement for ensuring the safe and stable operation of smart grids.

Although traditional manual inspection methods played a crucial role in ensuring the safe operation of transmission lines during the early stages of power grid development, their limitations have become increasingly evident with the continuous expansion of the grid and the growing complexity of transmission networks. Manual inspections suffer from low efficiency, extended inspection cycles, and high operational and maintenance costs, which hinder the realization of real-time and comprehensive equipment condition monitoring. Traditional manual inspection typically requires two inspectors to spend 2–3 days to conduct a comprehensive patrol of a 50-km transmission line, with a routine inspection cycle of once per month. Moreover, the quality of inspections heavily relies on the professional expertise and subjective judgment of personnel, making the process prone to missed detections, misjudgments, and delayed fault identification due to oversight, varying skill levels, or fatigue. Additionally, inspection tasks frequently require personnel to work in challenging environments such as complex terrain, adverse weather, or at elevated heights, posing serious safety risks including falls, electric shocks, and exposure to hazardous outdoor conditions. These factors significantly undermine the sustainability and reliability of traditional inspection approaches.

In recent years, with the rapid advancement of deep power vision technologies, intelligent inspection of transmission lines has emerged as a key focus in both academic research and industrial applications. Compared with traditional manual inspection methods, intelligent inspection significantly enhances the efficiency and reliability of transmission line operation and maintenance through automation, informatization, and intelligent technologies. By leveraging advanced equipment such as unmanned aerial vehicles (UAVs), inspection robots, high-resolution cameras, infrared thermal imaging, and LiDAR, intelligent inspection systems enable high-precision, long-range, and all-weather monitoring of transmission lines and their associated components, effectively overcoming the limitations of manual inspections in complex terrains and harsh environments. UAV-based inspection can complete a rapid scan of an equivalent-length line within 2 h, and the inspection frequency can be increased to weekly or even daily, resulting in an efficiency improvement of approximately 15–30 times compared to manual inspection. Furthermore, integrated with artificial intelligence and image recognition algorithms, these systems can automatically detect common defects such as broken wire strands, damaged insulators, and foreign object intrusions, thereby reducing errors caused by human judgment and improving the timeliness and accuracy of fault detection. In terms of fault response, intelligent monitoring systems can achieve second-level fault warnings, whereas the average time from fault occurrence to detection by manual inspection may range from several hours to several days, significantly improving the timeliness of fault detection. In addition, intelligent inspection systems offer data storage and historical comparison capabilities, enabling trend analysis and fault prediction to support predictive maintenance of the power grid. While ensuring personnel safety, intelligent inspection also facilitates reduced-manpower or even fully unmanned inspection processes, serving as a critical enabler for the transformation of power systems toward intelligent operation and maintenance.

Intelligent inspection is playing an increasingly important role in wire defect detection tasks. Leveraging advanced sensing equipment such as high-resolution cameras, infrared thermal imagers, and LiDAR, intelligent inspection systems enable comprehensive and detailed image acquisition of transmission wires. These systems can accurately identify common defects including strand breakage, strand unraveling, slackening, corrosion, and foreign object attachment. This significantly enhances the accuracy and efficiency of defect detection, effectively mitigating issues commonly associated with traditional manual inspection, such as a limited field of view and human oversight.

With the widespread adoption of inspection platforms such as intelligent gimbals and unmanned aerial vehicles (UAVs) in power line inspection, along with advancements in high-resolution imaging technologies, inspection images of transmission lines are increasingly characterized by large spatial coverage and high resolution. For the sake of clarity, this paper refers to such images—featuring extensive spatial scope and high information density—as large-format images. In large-format images, the background environment tends to be highly complex and variable, containing various interfering objects such as buildings, vegetation, towers, and other cables. These objects may share similar texture, color, or structural features with transmission wires, causing the wire regions to exhibit low contrast, blurred edges, or incomplete structures in the image. Such visual ambiguities severely hinder accurate wire detection and subsequent defect localization. This challenge is further exacerbated in aerial imaging scenarios, where variations in viewing angles, uneven lighting, occlusions, or motion blur may lead to false detections, missed detections, or failures in identifying wire discontinuities. Therefore, efficiently and reliably extracting wire regions from large-format images has become a critical issue in advancing the automation and intelligence of power line inspection.

At present, most mainstream wire detection methods are based on deep learning models, which rely heavily on large amounts of annotated data for supervised training. These approaches primarily include object detection and segmentation algorithms built on convolutional neural networks (CNNs) or Transformer-based architectures. While such methods generally perform well in standardized scenarios or benchmark datasets, they face several limitations in real-world inspection environments. First, the acquisition of large-scale annotated datasets is costly and time-consuming, often suffering from issues such as subjectivity and inconsistency in labeling. Second, both training and inference of deep models demand high-performance computing resources, making them difficult to deploy on embedded platforms or edge devices. Finally, deep learning models still exhibit limited generalization capabilities when confronted with highly complex backgrounds or images where wire structures are visually ambiguous. Traditional machine learning-based object detection methods offer several advantages, including strong interpretability, compatibility with small datasets, low computational requirements, simple implementation, and flexible deployment. However, they also suffer from notable drawbacks such as low detection accuracy, poor robustness, lack of end-to-end capability, and limited effectiveness in handling multi-class and multi-object scenarios. Therefore, there is significant practical value in developing a wire extraction method characterized by training-free operation, strong robustness and generalization, and ease of deployment in real inspection settings.

To address the aforementioned challenges, this paper proposes a training-free transmission wire extraction method specifically designed for large-format images with complex backgrounds. The method is based on computer vision techniques and consists of three main stages: (1) Single-image depth estimation is employed to enhance the structural perception of the image, thereby improving the visual separability between the foreground wires and the background; (2) Structural analysis techniques are used to extract elongated and continuous line segment candidates, and a line grouping strategy is applied to identify wire fragments; (3) Classical algorithms such as edge detection, morphological processing, and probabilistic Hough transform are integrated to accurately extract wire regions and eliminate redundant detections.

The main contributions of this paper are as follows:A detection mechanism that integrates depth estimation with structural line segment analysis is proposed to address the elongated structure and weak edge features of transmission wires in large-format images, effectively enhancing their visual representation second bullet.A complete visual processing pipeline is designed to extract wire regions without relying on any model training, demonstrating strong engineering applicability.Systematic experiments are conducted on a real-world transmission line inspection image dataset, validating the effectiveness and superior performance of the proposed method in complex background scenarios.

The structure of this paper is organized as follows: Section 2 reviews the current research status in the field of wire detection and relevant deep learning-based vision techniques. Section 3 presents a detailed description of the overall framework and key technical components of the proposed method. Section 4 evaluates the performance of the method through experiments and provides a comparative analysis with mainstream detection approaches. Section 5 concludes the paper and discusses future research directions and potential engineering applications.

## 2. Related Works

In recent years, rapid advancements in image processing algorithms have driven continuous improvements in wire detection technologies. Object detection methods based on traditional machine learning algorithms have been widely applied in industrial defect detection. These methods typically rely on low-level image features such as edges, textures, and grayscale distributions, combined with techniques including threshold segmentation, edge detection, morphological operations, Hough transform, and classical feature matching algorithms like SIFT and SURF to achieve wire extraction and defect localization. Due to their relative simplicity and high computational efficiency, these algorithms played a significant role in early wire detection systems and are particularly suitable for scenarios with clear structures and minimal background interference. However, traditional machine learning methods often suffer from limited robustness and high false detection rates when confronted with complex backgrounds, varying illumination conditions, or wire deformations. Nonetheless, traditional approaches still hold practical value in resource-constrained settings and can serve as effective preprocessing or auxiliary mechanisms to enhance the efficiency and stability of deep learning-based detection systems.

Currently, traditional machine learning-based object detection algorithms have made significant progress in industrial defect detection. Liu et al. [1] developed an intelligent wire installation quality inspection robot integrated with image recognition capabilities, employing the SIFT algorithm to extract texture features of standard wires and identify defective abnormal regions. Yang et al. [2] utilized airborne LiDAR point cloud data combined with an improved Hough transform and RANSAC fitting to achieve rapid extraction of wires in high-voltage transmission line point clouds. Zhu et al. [3] proposed a detection method for energized transmission line operation robots based on an improved Hough transform. Song et al. [4] introduced an enhanced randomized Hough method combining Hessian matrix preprocessing, boundary search, and pixel row segmentation, enabling faster and more accurate transmission line detection in UAV images with complex backgrounds and significantly reducing false detection rates. Ye et al. [5] presented an improved approach integrating the Canny operator and Hough transform, which better extracts the edges and abnormal parts of the target imager while filtering out unwanted background noise points, effectively detecting tower corrosion. Luo et al. [6] proposed a power line recognition algorithm under complex ground object backgrounds, using an improved Ratio operator combined with contour feature-based background denoising for edge detection, and applying a Hough transform-based dynamic grouping and filtering of lines to extract edges and ultimately identify complete power lines. Zou et al. [7] employed fusion of RGB and NIR images, first extracting candidate regions through edge detection and constructing thin-line structural constraints, then combining near-infrared intensity verification to substantially improve power line segmentation accuracy. Liu [8] adopted an improved K-means segmentation combined with Otsu thresholding for binarizing grayscale images in transmission line strand break and damage detection, followed by morphological operations for noise removal and smoothing, significantly enhancing wire region extraction accuracy. Wang et al. [9] proposed a detection method for transmission line strand breaks and foreign object defects based on line structure awareness, employing horizontal and vertical gradient operators capable of detecting line width to extract linear objects in inspection images, followed by analysis of collinearity, proximity, and continuity laws to connect discontinuous segments into long lines, and identifying significant parallel wire groups via parallelism calculations. Tong et al. [10] developed an algorithm for automatic extraction and recognition of transmission lines from natural complex backgrounds in aerial images, using a Marr-Hildreth edge detection algorithm to enhance linear features, Hough transform to extract lines and curves, and morphological analysis to differentiate transmission lines from background objects.

Deep learning-based object detection algorithms have also achieved significant advances in industrial defect detection tasks. Xu et al. [11] proposed an improved YOLOv3 rapid detection method that optimizes object candidate boxes through clustering analysis. Compared with Faster R-CNN and SSD, their method substantially increases detection speed while maintaining accuracy. Huang et al. [12] developed a lightweight object detection framework, TLI-YOLOv5, tailored for transmission line inspection tasks, addressing the demanding requirements of large-scale modern transmission line monitoring and providing a reliable and efficient solution. Wang et al. [13] proposed a transmission line detection algorithm based on adaptive Canny edge detection to address the difficulty of identifying transmission lines from complex ground environments during the autonomous inspection process of transmission line inspection robots. This algorithm has a high recognition rate under different ground backgrounds and heights from the wire, which can meet the real-time detection requirements of the inspection robot wire. Wang et al. [14] proposed an improved algorithm, Line-YOLO based on YOLOv8s-seg, capable of efficiently performing wire angle detection tasks. Target detection methods based on depth estimation algorithms have also made notable progress in industrial defect detection. Jiang et al. [15] conducted research on YOLO algorithm and monocular visual wire detection technology for four types of wires: LGJ-120/25, LGJ-300/40, LGJ-400/50, and LGJ-630/45 By optimizing the YOLO algorithm to detect wire texture, using monocular vision to measure wire width, and integrating the two to determine wire model, we aim to develop a precise and efficient wire model recognition system. Yang et al. [16] proposed a transmission line detection model based on the YOLO algorithm for the detection of transmission lines in visible-light images captured by, e.g., drones. Jing et al. [17] proposed a detection method for exposed conductors in 10 kV distribution lines based on an improved YOLOv8 algorithm, to assist power grid operation and maintenance personnel in quickly and efficiently detecting exposed conductor defects. Yi et al. [18] proposed an improved foreign object detection method, SC-YOLO, for transmission lines using YOLOv8, which enhances the model’s ability to dynamically adjust to different inputs and focus on key information. Liu et al. [19] conducted research on infrared images of overhead transmission lines collected by drones and proposed an infrared image segmentation method based on Hough line detection, which achieved ideal segmentation results for the wire area in the infrared image. Lin et al. [20] proposed a multi-objective detection and localization method for transmission line detection images based on an improved Faster RCNN (Faster Region Convolutional Neural Network), which improved the detection speed and established an improved Faster RCNN model suitable for detecting polymorphic features in images. Vemula et al. [21] proposed using the deep learning Resnet50 architecture as the backbone network in power line detection systems, combined with the feature pyramid network (FPN) architecture for feature extraction. The Regional Proposal Network (RPN) is trained end-to-end to create regional recommendations for each feature map. Jin et al. [22] designed a transmission line anomaly detection method based on an improved convolutional neural network Extract abnormal features of transmission lines, generate feature maps, and mark the location of line anomalies Constructing a line anomaly detection model based on improved convolutional neural networks, which reduces the redundancy of critical sampling samples for transmission lines, and performs line detection on transmission lines. Wang et al. [23] proposed a high-performance and lightweight substation defect detection model based on YOLOv5m, called YOLO Substation Large, which solves the problems of untimely defect detection and slow response in the intelligent inspection process. Dong et al. [24] proposed the PHAM-YOLO (Parallel Hybrid Attention Mechanism You Only Look Once) network for automatic detection of meter defects in substations.

Li et al. [25] proposed a depth map-based wire detection method that obtains monocular depth maps and geometric regression to automatically locate wire segments using a regression network, achieving high-precision localization. This approach accurately extracts wire centerlines and performs spatial position regression, effectively preserving structural integrity in complex scenarios. Hao et al. [26] studied a damage detection method for overhead transmission lines based on image processing by using drones to collect images of overhead transmission lines and applying image recognition processing technology to process the collected images, and the detection accuracy of wire cracks was improved.

In recent years, with the widespread adoption of UAV-based inspection technologies, several public datasets have been released, providing critical support for research on intelligent analysis of transmission lines. Among them, the Powerline dataset [27] is one of the earliest benchmark datasets designed for power line semantic segmentation. It contains a large number of UAV-captured transmission line images with pixel-level wire annotations, enabling the training and evaluation of semantic segmentation models. Building upon this, the UAV-Powerline dataset [28] further enhances data diversity by covering aerial images under various weather, illumination, and terrain conditions, and provides instance-level annotations, making it suitable for fine-grained tasks such as wire instance segmentation and topological structure analysis. Additionally, the PLD (Power Line Dataset) [29] focuses on wire extraction and geometric reconstruction from high-resolution airborne imagery, featuring higher image resolution and supporting automated processing and 3D modeling of long-distance transmission corridors. The release of these public datasets has significantly advanced the development of deep learning-based methods for wire detection and defect recognition, particularly by providing standardized benchmarks for supervised model training and comparative evaluation.

## 3. Proposed Methods

This paper proposes a wire region extraction method for transmission line images, with the overall processing pipeline illustrated in Figure 1. The method first performs depth estimation, where a depth estimation model is employed to generate a depth map from the original large-format images, thereby enhancing the distinction between the foreground (wires) and the background. Subsequently, edge detection and morphological operations are applied to achieve effective line-segment grouping. Finally, a cropping decision mechanism is employed to eliminate non-wire structures from the valid line-segment groups, ultimately enabling precise extraction of the wire regions.

Unlike existing methods that rely heavily on deep learning models, the proposed approach combines traditional computer vision techniques with depth estimation without requiring any training process. It features low computational overhead and is well-suited for engineering deployments in data-scarce or resource-constrained environments. While improving wire detection accuracy, this method significantly reduces dependence on large-scale annotated datasets and computational resources, offering a feasible training-free solution for efficient extraction of critical components in transmission line images.

### 3.1. A Depth Image-Guide Foreground-Background Separation Method

Under complex background conditions, power line regions are typically slender and exhibit color similarities with background elements such as buildings and vegetation, which causes traditional line structure extraction methods based solely on RGB images to be easily disturbed. To enhance the distinguishability between wire regions and the background, this paper introduces a foreground-background separation mechanism based on depth maps, leveraging spatial depth information of different image regions to structurally enhance and separate the wires.

Specifically, the original RGB image is first input to the open-source general monocular depth estimation model Depth Anything v2 to generate a depth map with the same resolution as the original image. The entire process is illustrated in Figure 2. This model demonstrates strong generalization capability and can output smooth and continuous relative depth maps without additional training, effectively characterizing the spatial structural features of various regions within the scene. In power inspection images, wires are usually distributed horizontally and suspended against distant backgrounds, thus appearing as continuous and smooth far-distance regions in the depth map.

After obtaining the depth map, the Sobel operator is applied to compute gradients and extract vertical edge response regions, emphasizing horizontal linear structures. To further enhance edge representation, the depth edge map is fused with the edge map extracted from the original image using the Canny operator. This fusion preserves strong edges while suppressing irrelevant texture noise. The fused edge map is then subjected to morphological dilation to connect broken line segments, thereby improving the coherence and completeness of linear targets. The entire procedure is depicted in Figure 2.

### 3.2. A Line Segment Structure-Based Method for Wire Extraction

In practical scenarios, wires in large-format images of transmission line corridors typically exhibit certain geometric and linear shape distribution characteristics, such as a consistent angular tilt, spanning roughly across the left and right sides of the image, similar angles, and considerable length. However, the images often contain numerous linear interfering elements, such as utility poles and rooftop edges, which pose significant challenges to accurate wire extraction. To address this, this paper proposes a wire region identification mechanism that combines line segment structural analysis with rule-based judgment to efficiently generate precise cropping regions. This mechanism consists of two key steps, namely line grouping and cropping decision. The visualization flowchart of the proposed method is shown in Figure 3.

#### 3.2.1. Line Grouping

After obtaining initial line segments via edge detection and Hough line transform, the system performs clustering based on geometric features including line segment orientation, length, and spatial relationships, grouping line segments that potentially belong to the same wire. After line grouping, invalid (non-wire) line-segment masks in the edge-detected image can be removed, while valid (wire) line-segment masks are retained. As illustrated in Figure 4, the red rectangular boxes denote valid line-segment groups that are preserved after the grouping step, whereas the circular boxes denote invalid line-segment groups that are removed through this process.

The specific process is to first use a probabilistic Hough transform to detect a set of line segments from the depth map, denoted as $L={\left\{l_{i}\right\}}_{i=1}^{N}$. Each line segment $l_{i}$ is represented by its two endpoint coordinates as $\left(x_{i1},y_{i1}\right)$ and $\left(x_{i2},y_{i2}\right)$. Then, the direction angle $\theta_{i}$ and length $L_{i}$ of each line segment are calculated, defined as follows:$$\theta_{i}=\arctan2(y_{i2}-y_{i1},x_{i2}-x_{i1})$$$$L_{i}=\sqrt{{\left(x_{i2}-x_{i1}\right)}^{2}+{\left(y_{i2}-y_{i1}\right)}^{2}}$$

Among them, the arctan2 function returns the angle between the line segment and the horizontal axis, with an angle range of $\left[{-180}^{{}^{\circ}},180^{{}^{\circ}}\right]$, and the length $L_{i}$ represents the Euclidean distance of the line segment.

To ensure consistency in the direction of line segments within the group, the segments are clustered and grouped based on their angular similarity. Specifically, define the angle threshold $\tau_{\theta }$; if the average angle $\bar{\theta_{j}}$ between line segment $l_{i}$ and all line segments in the existing group $G_{j}$ satisfies:$$|\theta_{i}-\bar{\theta_{j}}|<\tau_{\theta }$$
then merge group $l_{i}$ into group $G_{j}$; otherwise, create a new group. In this article, the angle threshold $\tau_{\theta }$ is set at 5° to maintain strong directional consistency.

For each group $G_{j}$, further calculate its horizontal span:$$W_{j}=\underset{l\in G_{j}}{max}\;max(x_{l1},x_{l2})-\underset{l\in G_{j}}{min}\;min(x_{l1},x_{l2})$$

Simultaneously calculate the standard deviation of the angle of the group of line segments:$$\sigma_{\theta ,j}=\sqrt{\frac{1}{\left|G_{j}\right|}\sum_{l\in G_{j}}{\left(\theta_{l}-\bar{\theta_{j}}\right)}^{2}}$$

Compare the span $W_{j}$ with the image width $W_{img}$ and set a threshold ratio α (with a value of 2/3); if it satisfies:$$W_{j}>\alpha \times W_{\mathrm{img}}$$
and the standard deviation of the angle meets the requirements:$$\sigma_{\theta ,j}<\tau_{\sigma }$$
where $\tau_{\sigma }=5^{\circ }$, then it is determined that the group is a valid line segment group. This judgment effectively eliminates line segment groups with insufficient length and inconsistent direction, thereby filtering out background and noise interference.

#### 3.2.2. Cropping Decision

To improve the accuracy of wire region cropping, this paper further proposes a filtering-based cropping decision mechanism (referred to as the wire extraction model). This mechanism is designed to eliminate irrelevant features from the initially detected line segment regions, ensuring that the final cropped results exhibit strong structural coherence and discriminative capability. The core rationale behind this mechanism is based on a geometric prior: wires typically appear as long, straight structures that traverse the entire image and are expected to intersect with the image boundaries. Based on this observation, a two-step decision process is designed. First, structural mask regions are constructed based on grouped line segments; second, boundary-contact analysis is performed on these mask regions to determine whether they should be retained for subsequent cropping. For each clustered line segment, denoted as $G=\left\{l_{1},l_{2},\ldots ,l_{n}\right\}$, it is plotted as a thick line with a width of w on the image to construct a binary mask map M. Each line segment $l_{i}$ is rendered as a set of pixels, and the mask regions corresponding to multiple line segments are merged to form the overall region M. Subsequently, a morphological dilation operation is performed on the mask map M to enhance the connectivity between line segments; Further corrosion operation is carried out to suppress isolated noise, ultimately obtaining the structural mask region M’. Perform connected region analysis on the processed mask M’ and extract all non-intersecting regions, denoted as $R_{1},R_{2},{\ldots ,R}_{m}$. For each region $R_{i}$, calculate its minimum bounding box $B_{i}$, denoted as $\left(x_{i},y_{i},w_{i},h_{i}\right)$, where $\left(x_{i},y_{i}\right)$ is the upper left corner coordinate, and $w_{i}$ and $h_{i}$ are the width and height, respectively. Let the total width and height of the image be W and H, respectively, and define an edge contact tolerance δ (where δ is set to 10 pixels in this article). If the bounding box of a region satisfies any of the following conditions, it is considered to be in contact with the edge of the image: (1) $x_{i}\leq \delta$, represents contact with the left boundary of the image (2) $x_{i}+w_{i}\geq W-\delta$, represents contact with the right boundary (3) $y_{i}\leq \delta$, represents contact with the upper boundary (4) $y_{i}+h_{i}\geq H-\delta$, represents contact with the lower boundary.

In practical applications, only areas that meet the requirements of (1) simultaneously contacting the left and right boundaries, (2) simultaneously contacting the left and upper boundaries, and (3) simultaneously contacting the lower and right boundaries are considered effective wire areas. Record all connected regions that meet the judgment rule as R. The effectiveness of the judgment mechanism is illustrated in Figure 5. In the left image, the line-segment masks enclosed within the red rectangular boxes represent those that meet the criteria and are retained after the cropping judgment mechanism. In contrast, the line-segment masks enclosed within the elliptical boxes do not satisfy the cropping judgment mechanism and are therefore removed.

Finally, the system will crop the image based on the position corresponding to R in the original image, extracting the wire target area with continuity and structure. This mechanism can effectively eliminate complex interference factors such as house edges, tree branches, and background textures, significantly improving the accuracy and discriminability of cropping results, and enhancing the overall system’s generalization ability in multiple scenarios.

## 4. Experiments

### 4.1. Experimental Setup

The experiments were conducted on a Linux-based Ubuntu 20.04.4 LTS operating system using an NVIDIA Tesla P100 GPU. All experiments were performed on three datasets. Firstly, two categories of images are defined according to the background complexity: (1) simple-background images (with a background consisting of a single sky or a single ground surface), and (2) complex-background images (with backgrounds containing numerous linear structures such as farmland or buildings), as shown in Figure 6. Here, the three datasets are referred to as Dataset A (short for DA), Dataset B (short for DB), and Dataset C (short for DC). Dataset A comprises 230 transmission line images, including 180 images containing wires (90 with simple backgrounds and 90 with complex backgrounds), and the remaining 50 images do not contain wires. Dataset B consists of 90 simple-background wire images and 50 wire-free images from Dataset A. Dataset C includes 90 complex-background wire images from Dataset A and the same 50 wire-free images. This setup facilitates the comparison of image recognition performance under different background conditions. The input image resolution is 12,768 × 9564 pixels, requiring no additional preprocessing. Quantitative evaluation metrics are described in Section 4.2.

### 4.2. Evaluation Metrics

To systematically evaluate the effectiveness of the proposed wire extraction method in practical large-format image processing tasks, this study establishes a quantitative evaluation metric system based on image cropping performance. The evaluation focuses on the method’s performance across different image types and varying levels of background complexity, and is primarily centered around the following three core metrics:
1.OCA (Overall Cropping Accuracy)

OCA is used to measure the algorithm’s ability to successfully locate and crop areas containing wires in all test images. The specific definition is:(1)$$OCA=\frac{I_{crop\_line}+I_{no\_crop\_line}}{I_{Total\;images}}$$

Among them, $I_{crop\_line}$ is the number of images that successfully cut wires in the image group containing wires. $I_{no\_crop\_line}$ is the number of images that have not been cropped out wires in the image group without wires. $I_{Total\;Images}$ is the number of all images.

2.WICA (Wire Image Cropping Accuracy)

WICA is used to measure the algorithm’s ability to successfully locate and crop areas containing wires in test images that only contain wires. The specific definition is:(2)$$WICA=\frac{I_{crop\_line}}{I_{with\_line}}$$

In this formula, $I_{with\_line}$ is the number of images containing wires.

The industrial dataset used for evaluation was collected without pixel-level annotation due to cost and scalability constraints. Therefore, we conducted the evaluation through manual verification on the final output—that is, human experts inspected each cropped result and the original images to determine whether the localization and cropping decisions were correct. This process allowed us to compute OCA and WICA reliably, even in the absence of segmentation masks.

### 4.3. Comparison with Deep Learning Methods

To validate the effectiveness of the proposed method, this study conducted comparative experiments against several state-of-the-art object detection algorithms. Multiple representative lightweight YOLO variants (e.g., YOLOv8n, YOLOv9t, YOLOv10n, and YOLOv11n) were selected as baseline models to ensure a fair comparison in terms of computational efficiency and deployment feasibility. All methods were evaluated under identical datasets and experimental settings to ensure a fair performance comparison. The evaluation metrics included Overall Classification Accuracy (OCA) and Wire-Image Classification Accuracy (WICA).

The experimental results, as shown in Table 1, indicate that our method achieves an OCA of 98%, representing an improvement of approximately 5.1 percentage points over the best-performing YOLOv11n (92.9%) and 14.5 percentage points over the lowest-performing YOLOv10n (83.5%). In terms of the WICA metric, our method similarly achieves a remarkable score of 97.78%, outperforming all other models, with the largest gain reaching 10.58 percentage points compared to YOLOv11n (87.2%). These results demonstrate that our approach not only offers superior overall detection accuracy but also maintains exceptionally high precision when processing complex images containing wires, reflecting strong robustness and generalization capability. In contrast, the YOLO series models exhibit noticeable performance gaps in both metrics, indicating that our method has better adaptability and practical value for specific task scenarios.

### 4.4. Comparison with Traditional Experimental Methods

To further validate the proposed method’s robustness in complex backgrounds and its ability to control false detections, three representative traditional image processing methods were selected as baseline comparisons. These included: the Canny + Hough line detection method, the Sobel + connected-component analysis approach, and a morphological skeleton-based extraction method. All methods were applied to the same input images and followed a unified cropping decision strategy. Final cropping results were manually annotated for evaluation, with two quantitative metrics calculated: Overall Classification Accuracy (OCA) and Wire-Image Classification Accuracy (WICA).

As shown in Table 2, our method significantly outperforms traditional image processing baselines in both evaluation metrics. For OCA, our method achieves 98%, representing improvements of 26.5 and 25.0 percentage points over Canny + Hough (71.5%) and Sobel + Connected Components (73%), respectively. For WICA, our method attains 97.78%, with corresponding gains of 29.48 and 27.78 percentage points compared to the two baselines (68.3% and 70%). The improvement is particularly pronounced in the WICA metric, indicating that in complex scenarios containing wires—characterized by fine linear structures, low contrast, occlusions, and background texture interference—the proposed method can localize targets more stably while effectively suppressing false detections.

### 4.5. Ablation Experiment

To further assess the specific contributions of each key component to the overall performance of the proposed wire extraction method, an ablation study was conducted based on a modular decomposition strategy. Specifically, four representative submodules within the processing pipeline were selected for evaluation: Depth Estimation (DE), Line Grouping (LG), Region Judgment (RJ), and Morphological Processing (MP). In each experimental group, only one module was removed while keeping the rest of the pipeline unchanged, in order to observe the resulting performance variations. Specifically, our method incorporates all modules; Method A1 excludes the Depth Estimation (DE) module; Method A2 excludes the Line Grouping (LG) module; Method A3 excludes the Region Judgement (RJ) module; and Method A4 excludes the Morphological Processing (MP) module. All experiments were conducted under consistent conditions, using the same dataset, and final cropping results were manually annotated for evaluation. The assessment metrics included Overall Cropping Accuracy (OCA) and Wire-Image Cropping Accuracy (WICA).

As shown in Table 3, removing any individual module leads to a performance drop, demonstrating that all four submodules contribute positively to the overall system. Removing Depth Estimation (A1) lowers the performance to 94.7%/95%, a decrease of 3.3 and 2.78 percentage points, indicating that depth priors can provide beneficial foreground–background separation in scenarios involving occlusion and background clutter. The absence of Line Grouping (A2) also causes notable degradation (90.0%/88.9%), with reductions of 8.0 and 8.88 percentage points, suggesting that this module effectively consolidates fragmented edges and enhances the directional continuity of wires, which is crucial for stable cropping. The Region Judgment has the greatest impact on performance: after its removal (A3), OCA/WICA decrease to 85.5%/83.39%, representing drops of 12.5/14.39 percentage points compared with the complete model (98%/97.78%). This indicates that performing region-level filtering under complex backgrounds can significantly suppress false detections of non-wire structures, which is critical for improving WICA. The absence of Morphological Processing (A4) results in 95.5%/95%, with declines of 2.5 and 2.78 percentage points, implying that this module primarily refines results and repairs local defects, offering stable but relatively modest gains in accuracy.

Our method achieves the highest performance of 98%/97.78%, validating the robustness and generalization capability of the proposed modular design in complex scenarios. We further conducted experiments on two additional datasets (DB and DC), as detailed in Appendix A.

### 4.6. Comparison of Processing Time Experiment

To evaluate the processing efficiency of the proposed method in real-world deployment scenarios, five deep learning-based detection methods (including YOLOv8n and YOLOv9n) and two traditional image processing methods were selected for comparison. The average processing time per image was measured across the entire pipeline—from image input to final wire extraction—including both the depth estimation stage (using a pre-trained model) and the subsequent geometric filtering stages. All experiments were conducted on the same platform (NVIDIA Tesla P100, 16GB VRAM). Deep learning models were executed on a GPU. For fair comparison, the proposed method’s post-processing steps were implemented using GPU-accelerated libraries (e.g., CUDA-accelerated OpenCV), ensuring consistent hardware utilization. As shown in Table 4, the proposed method achieved an average processing speed of just 120 ms per image without relying on any model training, significantly outperforming the deep learning-based approaches. These results demonstrate the method’s high practical efficiency and strong suitability for deployment in resource-constrained environments, which is critical for robust IoT systems [37].

### 4.7. Visualized Experimental Results

To provide a more intuitive demonstration of the effectiveness and superiority of the proposed method, visualization experiments were conducted. Figure 7 presents a comparative visualization between our method and YOLOv10n. As can be clearly observed, the proposed method surpasses the baseline in terms of detail preservation, boundary segmentation, and semantic consistency. In particular, under complex backgrounds, our method achieves more accurate localization of target regions, whereas YOLOv10n is more susceptible to background interference, leading to frequent missed detections.

## 5. Discussion and Conclusions

To validate the effectiveness of the proposed training-free wire extraction method for large-format images of transmission line spans, a series of comparative experiments were designed and conducted. The experimental procedure consisted of three main steps: First, the proposed method was applied to a designated validation dataset to extract wires, and relevant performance metrics were recorded. Second, standard object detection models (e.g., YOLO variants) and conventional image processing methods were applied to the same dataset for wire detection, and their performance metrics were documented. Finally, a comparative analysis of the metrics across the three approaches was conducted. The results demonstrate that the proposed method outperforms both deep learning and traditional approaches across multiple metrics, confirming its robustness and accuracy in handling complex backgrounds. This underscores its high practical value and potential for supporting defect detection tasks in transmission line span images.

Unlike conventional approaches that rely heavily on supervised training, the proposed method integrates depth estimation to guide wire extraction in visually complex scenes. Specifically, the method involves the following steps: (1) obtaining a depth map of the input image using a depth estimation algorithm; (2) applying edge detection to extract candidate linear structures; and (3) employing a two-stage filtering strategy based on the proposed wire extraction model to remove non-target lines. This enables the accurate identification of wires even in cluttered backgrounds. By effectively enhancing the distinction between foreground and background regions, the method achieves reliable wire extraction without any training process. It provides a solid foundation for subsequent defect analysis and offers significant contributions toward improving the efficiency of power system maintenance and the automation level of image analysis.

Although the proposed training-free wire extraction method demonstrates strong adaptability and accuracy in large-format transmission line archive images, there remains room for further research and optimization. To provide deeper insight, we analyze representative failure cases observed in highly cluttered environments—such as those with dense vegetation, heavy shadows, or partial occlusions. Failure Analysis: In scenes with dense vegetation, the depth map often fails to distinguish between thin branches and actual wires due to similar depth values and texture confusion, leading to false positives or broken wire segments. This suggests that depth estimation inaccuracies are the primary bottleneck in such scenarios. In contrast, under strong shadows or backlighting, while the depth map remains relatively reliable, the edge detection stage may produce fragmented or discontinuous wire responses. In these cases, the subsequent line-grouping heuristics (e.g., span ratio, contact tolerance) struggle to reconnect broken segments, indicating limitations in the geometric reasoning module. Future work may focus on the following directions: (1) Improvement of Depth Estimation Accuracy: The quality of depth estimation currently has a direct impact on the accuracy of wire extraction. Future work could explore higher-resolution or structure-aware depth estimation to enhance depth map stability and detail preservation, especially at edges. (2) Enhanced Adaptability to Complex Scenes: In highly cluttered environments—such as those featuring vegetation, transmission towers, or shadows—the proposed method may still encounter local extraction failures. To improve robustness under extreme conditions, future research could incorporate multimodal information (e.g., infrared imagery, thermal imaging, or historical aerial survey data) to further strengthen scene understanding and extraction reliability.

## Figures and Tables

**Figure 1 sensors-25-06636-f001:**
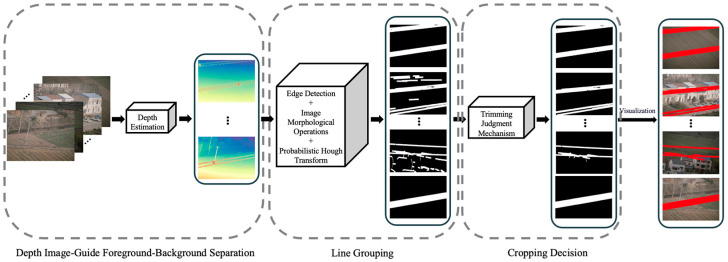
Overall method framework.

**Figure 2 sensors-25-06636-f002:**
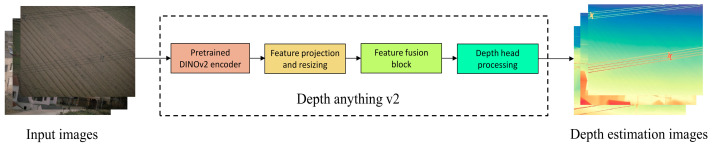
Depth Image-Guide Foreground-Background Separation.

**Figure 3 sensors-25-06636-f003:**
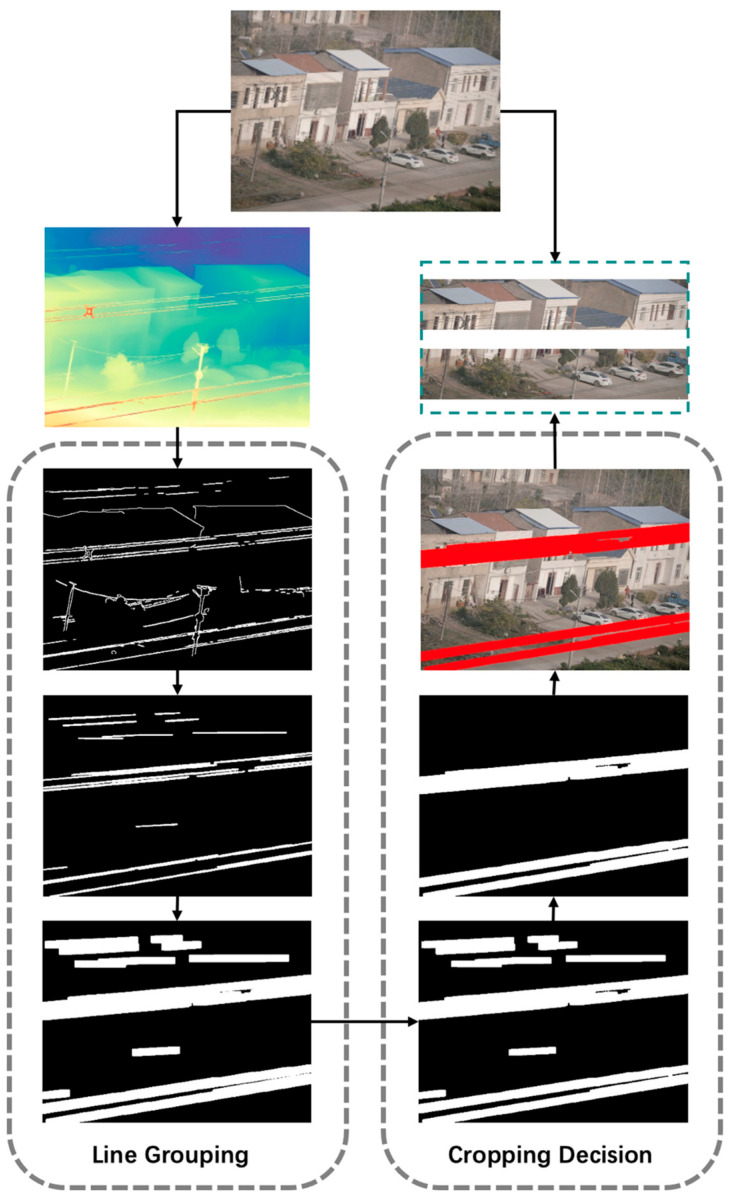
Line Segment Structure-Based Method for Wire Extraction.

**Figure 4 sensors-25-06636-f004:**
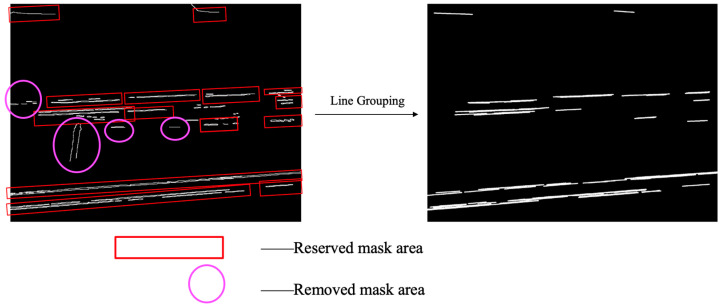
Invalid Line Segment Removal Process.

**Figure 5 sensors-25-06636-f005:**
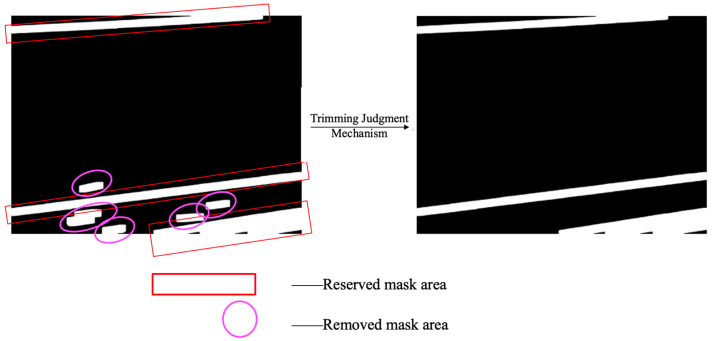
Edge Contact Filtering Process.

**Figure 6 sensors-25-06636-f006:**
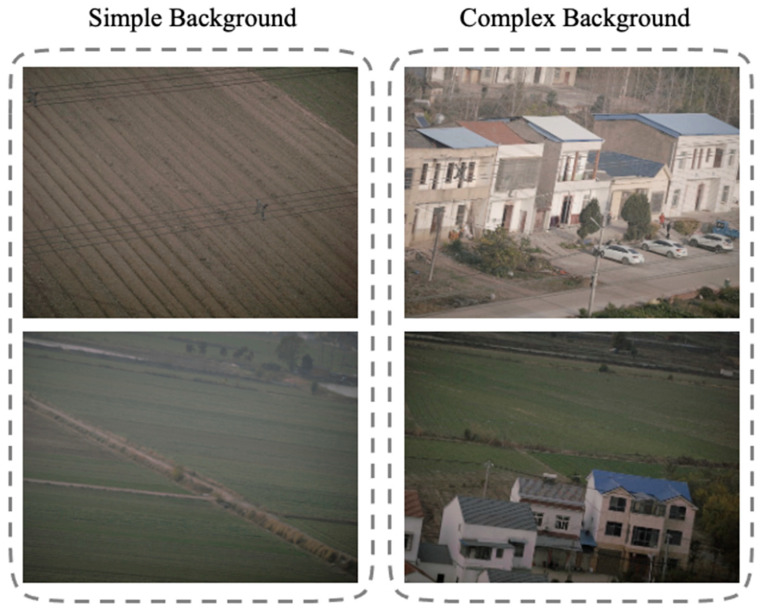
Dataset Category.

**Figure 7 sensors-25-06636-f007:**
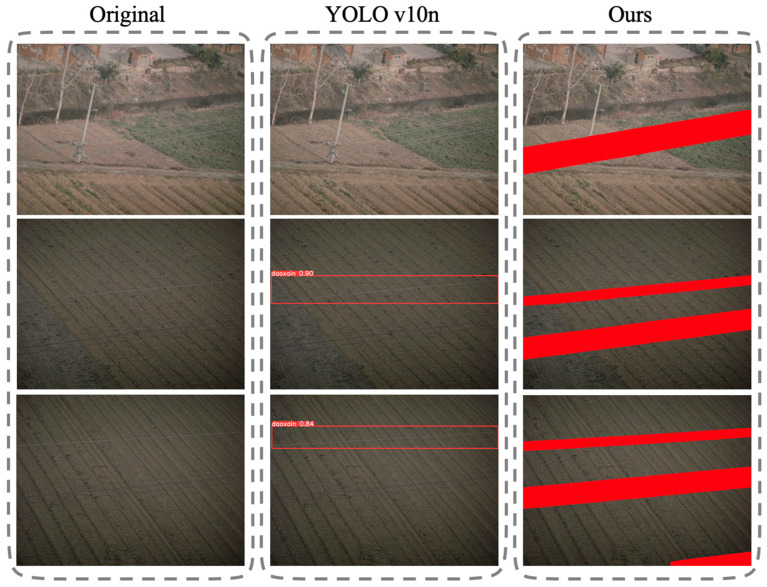
Visualized Experiment Result.

**Table 1 sensors-25-06636-t001:** Comparative experimental results of various models.

Model	OCA (DA)	OCA (DB)	OCA (DC)	WICA (DA)	WICA (DB)	WICA (DC)
YOLOv8n [30] YOLOv8m [30]	89.4% 95.5%	89.5% 91.8%	89.3% 89.7%	88.3% 88.5%	84.2% 86.1%	83.3% 86.1%
YOLOv9t [31]	87.9%	91.1%	86.4%	87.7%	86.7%	86.4%
YOLOv10n [32]	83.5%	92.1%	84.2%	89.4%	87%	75.6%
YOLOv11n [33]	92.9%	88.6%	86.4%	87.2%	82.2%	78.9%
**Ours**	**98%**	**98.57%**	**97.14%**	**97.78%**	**97.78%**	**95.56%**

**Table 2 sensors-25-06636-t002:** Comparison with traditional experimental methods.

Model	OCA (DA)	OCA (DB)	OCA (DC)	WICA (DA)	WICA (DB)	WICA (DC)
Canny + Hough [34,35]	71.5%	72.14%	60.71%	68.3%	56.67%	38.89%
Sobel + connected component [36]	73%	86.43%	88.57%	70%	78.89%	82.22%
**Ours**	**98%**	**98.57%**	**97.14%**	**97.78%**	**97.78%**	**95.56%**

**Table 3 sensors-25-06636-t003:** Results of ablation experiment on Dataset A.

Method	Depth Estimation	Line Grouping	Region Judgment	Morphological Processing	OCA (DA)	WICA (DA)
A1	×	√	√	√	94.7%	95%
A2	√	×	√	√	90.0%	88.9%
A3	√	√	×	√	85.5%	83.39%
A4	√	√	√	×	95.5%	95%
**Ours**	√	√	√	√	**98%**	**97.78%**

**Table 4 sensors-25-06636-t004:** Results of time comparison experiment.

Model	Total Processing Time	Pre-Training
YOLOv8n [30]	1200 s	√
YOLOv9t [31]	952.4 s	√
YOLOv10n [32]	833 s	√
YOLOv11n [33]	903.9 s	√
Canny + Hough [34,35]	749.02 s	×
Sobel + connected component [36]	772.56 s	×
**Ours**	**373.86 s+**	×

## Data Availability

The datasets presented in this article are not readily available because the data in this article belong to the State Grid Electric Power Space Technology Company Limited and were used in the Science and Technology project of the State Grid Electric Power Space Technology Company Limited. Due to the confidentiality of the project and other reasons, it is not possible to directly access the datasets. Requests to access the datasets should be directed to the corresponding author: baiyuncan1@163.com.

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
