# Peer review of "A Training-Free Foreground–Background Separation-Based Wire Extraction Method for Large-Format Transmission Line Images"

_sensors, 2025, doi:10.3390/s25216636_

Round 1

Reviewer 1 Report

Comments and Suggestions for Authors

Dear Authors,

Thank you for such important and relevant research in the field of transmission line image analysis. Overall, I agree with the structure of your manuscript regarding the relevance of the problem and the evidence base, but I have a couple of additional questions in the presentation itself, which I hope future editors will help your future readers understand the research in more detail:

1. Introduction. When discussing advantages and disadvantages, for example, "Manual inspections are often characterized by low efficiency, long inspection cycles, and high operational and maintenance costs, making it difficult to achieve real-time and comprehensive monitoring of equipment conditions" or "thereby reducing errors caused by human judgment
and improving the timeliness and accuracy of fault detection," be sure to provide numerical values ​​for the parameters so that the reader can understand the level of technology being discussed.

2. Section 3.2.2. Is there noise from wire vibrations under mechanical stress, such as from ambient wind? What error does this introduce into the analysis?

I wish you success in all your future research!
Reviewer

Reviewer 2 Report

Comments and Suggestions for Authors

This paper presents a training-free wire extraction method for large-format transmission line images with complex backgrounds. Unlike deep learning methods. It effectively separates slender wires from cluttered backgrounds and achieves robust performance without training. However there are some suggestions before publication.

Suggestion 1: At “Accurately extracting the regions corresponding to transmission wires is essential for enhancing both the efficiency and accuracy of subsequent defect detection.” Please shorten the sentence and avoids redundancy.

Suggestion 2: At “The core idea involves leveraging depth estimation maps to improve the foreground–background separability, thereby facilitating the robust identification of slender wire structures under visually cluttered conditions.” Please make the sentence clearer.

Suggestion 3: At “Unlike deep learning-based methods, the proposed method requires no pre-trained models or annotated datasets.” Please make it more concise, while stressing independence from datasets and training.

Suggestion 4: At “Experimental results show that our method effectively addresses background complexity and computational overhead in large-scale transmission line images processing.” Please Improves grammar.
Suggestion 5: At “Manual inspections are often characterized by low efficiency, long inspection cycles, and high operational and maintenance costs, making it difficult to achieve real-time and comprehensive monitoring of equipment conditions.” Please  stronger academic tone.

Suggestion 6: At “There is significant practical value in developing a wire extraction method that does not rely on training, exhibits strong robustness and generalization, and is easy to deploy in real inspection settings.” Make this part put the qualities in parallel form, increasing readability.

Suggestion 7: “The results demonstrate that the proposed method outperforms both deep learning and traditional approaches in multiple evaluation metrics, confirming its robustness and accuracy in handling complex background imagery." Rephrases for conciseness and avoids repetition of “proposed method.

Suggestion 8: At “Future efforts may explore the use of higher-resolution or structure-aware enhanced depth estimation algorithms to improve the stability and detail preservation of depth maps, particularly around edge structures.” Please reduce length while keeping all technical meaning.

Suggestion 9: Add reference at “These results demonstrate the method’s high practical efficiency and strong suitability for deployment in resource-constrained environments.” Add citation (Zhao, H.; Yan, L.; Hou, Z.; et al. *Error Analysis Strategy for Long-term Correlated Network Systems.* IEEE Internet of Things Journal, 2025.) here to emphasize IoT system robustness.

Reviewer 3 Report

Comments and Suggestions for Authors

The research domain is the detection and cropping of image regions containing electrical power lines.

This paper proposes and investigates a technology for detecting regions containing electrical power lines. It combines the following steps into a single processing workflow: (1) obtaining a depth map of the input image using a depth estimation algorithm; (2) applying edge detection to extract potential linear structures; and (3) applying a two-stage filtering strategy based on the proposed wire extraction model to remove non-target lines.

The newness of the article is about combining known methods into a unitary processing pipeline that solves the problem in the best possible way.

To conduct experimental studies, a dataset of large-format images was made, some with wires and some without. For images containing electrical power lines, two categories were defined based on background complexity. Two basic metrics were used for quality assessment: OCA (Overall Cropping Accuracy) and WICA (Wire Image Cropping Accuracy). Comparisons were made with both neural network models from the YOLO group and representative traditional image processing methods. It is quite interesting the conducted study of the contribution of basic processing modules to the estimated cropping accuracy.

Experiments on comparative estimation of the total operating time of all the studied methods are presented.

The discussion and conclusions in the article correctly correspond to the to the conducted research.

The following comments and recommendations should be made regarding the content of the paper.

  1. Approximately half of the presented review of related works is devoted to the problems of detecting industrial defects and foreign objects on power transmission line wires. This does not quite correspond to the main research domain of the article. It is advisable to adjust the review by expanding the proportion of methods for detecting regions with wires.
  2. The review did not consider papers describing datasets for the target area of research.
  3. In the paper, the authorsstatethat the evaluation focuses on the method’s performance across different image types and varying levels of background complexity.For the images containing wires, two categories were defined according to back- ground complexity: (1) simple-background images, and (2) complex-background images. Butin the subsequent tables with results there are no columns where the effectiveness of the compared methods would be presented separately for these categories. It is advisable to expand the table 1, 2 and 3 by columns OCA and WICA for simple-background, OCA and WICA for complex-background.
  4. There is no justification for the selected metrics: OCA and WICA. Unfortunately, these metrics do not allow us to estimate the accuracy of the region selection. They only evaluate the fact that the method detected the presence of wires in the image. Why weren'tothermetricsused? For example, a metric IoU often used in similar studies.
  5. The OCA metric was defined incorrectly. If it is not specified in the OCA definition that Ino_crop_lineshould be evaluated in the group of images without wires and Icrop_line should be evaluated in the group of images containing wires, then OCA always equals 1.

Reviewer 4 Report

Comments and Suggestions for Authors

This manuscript proposes a training-free wire extraction method for large-format transmission line images. The approach combines a depth-estimation stage using Depth Anything v2 with traditional computer-vision techniques such as edge detection, morphological operations, and probabilistic Hough transform. I find the work solid and potentially impactful, though a few points could be clarified or expanded to strengthen the contribution further.

(1) The term training-free is repeatedly emphasized, but the method still depends on the Depth Anything v2 model, which itself is a deep neural network trained on large datasets. Please clarify the meaning of “training-free”: does it refer to the fact that your method does not require task-specific retraining, or that it does not use any trained model parameters? Consider clarifying this point early or revising the terminology to prevent confusion and make the contribution clearer.

(2) It is highly recommended to review the manuscript for untranslated terms (e.g., “连通域”) and ensure all figure captions, labels, and variable names are presented in English for international readability.

(3) The algorithm relies on several manually set parameters (e.g., angle threshold = 5°, span ratio = 2/3, contact tolerance = 10 pixels). How were these values determined? Were they tuned empirically or based on a sensitivity analysis? If tuned, please explain whether the same parameters generalize across datasets or require adjustment for different image conditions.

(4) The method is described as computationally efficient (120 ms per image), but Table 4 later reports total processing times of several hundred seconds. Please unify the measurement definition (e.g., per image vs. total batch) and clarify whether GPU acceleration was consistently used across all model comparisons.

(5) For the YOLO baselines, please indicate whether lightweight versions (e.g., “n”, “t”) were used to ensure fair runtime comparisons. If so, could larger versions (e.g., YOLOv8m/l) achieve higher accuracy with comparable runtime after optimization? Clarifying this would help readers interpret the reported performance gap.

(6) The Discussion section briefly mentions future directions but remains general. It would be helpful to analyze why specific failure cases occur (for instance, in scenes with vegetation, shadows, or occlusions) and whether these issues originate mainly from depth-map inaccuracies or from the line-grouping heuristics. This analysis would make the discussion more informative and insightful.

Round 2

Reviewer 2 Report

Comments and Suggestions for Authors

After carefully reviewing the manuscript and confirming that the previous issues have been addressed, I find the paper ready for publication.

Reviewer 3 Report

Comments and Suggestions for Authors

Thank you for your detailed responses to my comments.

I am completely satisfied with the changes made in response to my comments 1-3 and 5.

But additionally, I think the next important part of your response to my comment 4 should be included in section 4.2 of the article: “The industrial dataset used for evaluation was collected without pixel-level annotation due to cost and scalability constraints. Therefore, we conducted the evaluation through manual verification on the final output – that is, human experts inspected each cropped result and the original images to determine whether the localization and cropping decisions were correct. This process allowed us to compute OCA and WICA reliably, even in the absence of segmentation masks.”

Reviewer 4 Report

Comments and Suggestions for Authors

The authors have addressed my previous comments well. The revised manuscript is clearer and better organized, with improved explanations of the “training-free” concept, consistent terminology, and a more detailed discussion of failure cases. Although a brief sensitivity analysis could still add value, the paper is now scientifically solid and suitable for publication.
